# Toward Transgene-Free Transposon-Mediated Biological Mutagenesis for Plant Breeding

**DOI:** 10.3390/ijms242317054

**Published:** 2023-12-02

**Authors:** Ilya Kirov

**Affiliations:** 1All-Russia Research Institute of Agricultural Biotechnology, 127550 Moscow, Russia; kirovez@gmail.com; 2Moscow Institute of Physics and Technology, 141701 Dolgoprudny, Russia

**Keywords:** transposons, plants, mutagenesis, RNA-dependent DNA methylation, gene silencing

## Abstract

Genetic diversity is a key factor for plant breeding. The birth of novel genic and genomic variants is also crucial for plant adaptation in nature. Therefore, the genomes of almost all living organisms possess natural mutagenic mechanisms. Transposable elements (TEs) are a major mutagenic force driving genetic diversity in wild plants and modern crops. The relatively rare TE transposition activity during the thousand-year crop domestication process has led to the phenotypic diversity of many cultivated species. The utilization of TE mutagenesis by artificial and transient acceleration of their activity in a controlled mode is an attractive foundation for a novel type of mutagenesis called TE-mediated biological mutagenesis. Here, I focus on TEs as mutagenic sources for plant breeding and discuss existing and emerging transgene-free approaches for TE activation in plants. Furthermore, I also review the non-randomness of TE insertions in a plant genome and the molecular and epigenetic factors involved in shaping TE insertion preferences. Additionally, I discuss the molecular mechanisms that prevent TE transpositions in germline plant cells (e.g., meiocytes, pollen, egg and embryo cells, and shoot apical meristem), thereby reducing the chances of TE insertion inheritance. Knowledge of these mechanisms can expand the TE activation toolbox using novel gene targeting approaches. Finally, the challenges and future perspectives of plant populations with induced novel TE insertions (iTE plant collections) are discussed.

## 1. Introduction

Plant breeding relies on the genetic diversity available to breeders, which is used to generate new combinations of genes and traits. Therefore, methods and approaches for generating genetic diversity in crops are the cornerstones of the plant breeding process. Implication of wild relatives and different genotypes and varieties in crosses followed by the selection of the most desirable genotypes from the progeny is a common way to create genetic diversity. Another approach involves chemical (e.g., EMS treatment) and radiation mutagenesis [1]. Mutagenesis has led to the development of new crop varieties and genetic resources [1]. Mutated genetic populations have been created for many plant species, leading to tremendous progress in functional genomics. However, radiation mutagenesis has several disadvantages, including a low frequency of valuable mutations and challenging identification of the causative mutations. However, novel techniques for precise gene targeting (e.g., genome editing) have been developed and adapted for many crop species [2,3]. Therefore, the application of classical mutagenesis in plant breeding has significantly decreased in recent decades [1].

Mutagenesis also occurs in nature but at a significantly lower rate than that achieved by artificial mutagenesis. It is obvious that in contrast to animals, plants cannot escape stress factors and cannot conquer new ecological niches by long-distance physical movement. Therefore, the main strategy for plants to adapt to new environmental conditions and stress is to explore their natural genetic diversity. It is now clear that environmental stress may increase the rate of single-nucleotide mutations [4] as well as structural rearrangements, including transposable element (TE) movements [5]. In the last few decades, a list of studies have concluded that stress may stimulate TEs to transpose in the genome and to broaden genetic diversity [6,7,8]. Importantly, novel TE insertions (TEIs) are not randomly generated over the genome and may occur frequently in stress-responsive genes, providing connections between stress action and the occurrence of novel alleles of stress-responsive genes [9,10,11]. This indicates that TE transposition is an evolutionary mechanism that serves as a natural biological mutagen and an important source of genetic diversity for plant adaptation.

Interestingly, TE transposition is also a major source of crop diversity and has been widely used by plant breeders for hundreds of years [12,13]. The gene candidate approach revealed a number of cases in which TE transposition in or near the genes caused the origin of novel phenotypic variants in different species [6,10,13]. Modern genomics has provided tools to perform genome-wide analyses of the association between trait variation and TEIs. Indeed, recent genome-wide association studies (GWASs) exploiting small variants (for example, SNPs and InDels) and TEI polymorphisms (TIPs) have revealed the great impact of TIPs on crop domestication and breeding [13,14]. For example, Hopscotch TE insertion into the teosinte branched1 (tb1) gene is associated with increased apical dominance in modern maize [15]. Also, TE insertions were shown to play a key role in tomato domestication, causing changes in tomato leaf shape, scene, and fruit color [13]. Recently, the implication of TE insertions in the diversity of modern *Brassica rapa* morphotypes has been demonstrated [14]. Therefore, TEs can be considered one of the most effective biological mutagens that can be exploited by plant breeders. The most recent understanding of TE biology and TE suppression mechanisms (e.g., RNA-directed DNA methylation pathways and various small RNAs), combined with the most recent molecular tools (e.g., CRISPR-based technology and virus-induced gene silencing), could allow for the use of TE mutational potential in the control mode.

TE insertions may result in a broad spectrum of consequences at the genome, epigenetic, transcriptome, and phenotype levels [12,13,14,15,16,17,18]. Based on this, the idea that the controlled activation of native TEs in a plant genome can bring new TE-mediated mutagenesis technology to accelerate plant breeding has been proposed and discussed recently [11,12,19]. In recent years, the validity of this concept has been established through research on the model organism *Arabidopsis thaliana* [18,19,20,21,22], but studies on crops are still lagging behind.

Here, I review TEs as biological mutagens that can be applied to perform a novel type of mutagenesis: biological mutagenesis. I address the following questions regarding TE-based mutagenesis in crops: What can and how can TE transposition be triggered in plants? How uniform is TE mutagenesis across the genomic loci? How is the inheritance of novel TEIs possible in plants? Finally, I describe the emerging perspectives on using modern gene-targeting technology to build a foundation for TE-mediated crop mutagenesis.

## 2. Genotype-Mediated TE Transposition Reactivation

### 2.1. Heterologous Expression of TEs via Plant Transformation

In 1983, the first active plant transposon, Ac, was cloned and sequenced from maize [23]. This began the era of TE application for the mutagenesis of plants via genetic transformation. In initial reports, the active TEs of one species were transformed into the genomes of other species to overcome host silencing mechanisms [24]. Using this approach, the transformed TEs were kept active in the foreign genome until they were recognized by the host silencing system. For example, Tos17 (rice), Tnt1, and TtoI (tobacco) retrotransposons were involved in gene tagging experiments in rice, *Arabidopsis thaliana*, *Medicago truncatula*, and other plant species [24,25,26,27,28]. It is important to note that these TEs are cell-culture-inducible, and therefore their activity can be tightly controlled or reactivated by a new round of in vitro regeneration [28]. Although heterologous expression of TEs plays an important role in understanding the TE transposition process and its silencing, the application of this approach for plant breeding is challenging in many countries because of the GMO legislation. Therefore, alternative approaches are required to establish GMO-free, breeder-friendly methods for TE-mediated mutagenesis.

### 2.2. Implication of DNA-Methylation-Deficient Genotypes

In the plant genome, TE activity is strictly controlled at different levels, including transcription (DNA methylation and histone modifications), post-transcription (post-transcriptional gene silencing), translation (ribosome stalling [29]), and post-translation barriers [29,30,31,32]. Among them, RNA-dependent DNA methylation pathways (PolIV-RDR2 and PolII-RDR6) tend to control most long and autonomous plant transposons [33]. Because DNA methylation plays a central role in TE silencing, mutations in genes involved in DNA methylation initiation and maintenance were used to trigger TE activity. Such mutants were obtained by mutagenesis, although a naturally occurring mutation in wild-type plants has also been reported [6]. The plants produced by crossing mutant lines with wild-types plants were used to produced so-called epiRILs (epigenetic recombinant inbred lines) [34,35]. Two types of mutants are mostly used for epiRIL production: met1 and ddm1 [36,37]. EpiRILs have extensively been used to study TEs and the epigenetic inheritance and trait control in model plants [6,10,34,35,37]. It was shown that, for example, *Arabidopsis* epiRILs possess multiple insertions of some TE families including AtCOPIA93, which possess EVD, one of the most active Arabidopsis LTR retrotransposons [38,39]. However, the application of epiRILs to boost TE transposition and phenotypic diversity in crops has not been demonstrated (Figure 1).

### 2.3. Utilization of Mutagenic Plant Lines with Reactivated TEs

Since the work of Barbara McClintock, it was discovered that some maize lines may possess increased mutagenic TE activity [40,41]. In her early works, McClintock observed ‘bizarre’ phenotypes after crossing maize lines carrying a chromosome with a ruptured end. Such phenotypes include leaves possessing twin sectors and cells with abnormal chlorophyll development [41]. Subsequently, the sequence of the corresponding transposons (Ac/Ds) was determined using the waxy allele carrying the insertion [23]. These studies showed, for the first time, that TEs can be naturally activated and that the insertions can be inherited, originating new alleles of genes. Sequencing of the Ac/Ds elements paved the way for the faster identification of the mutants and genes affected by insertions [42]. Another early example is Robertson’s Mutator maize lines, discovered by Robertson in 1978 [40] (Figure 1). Mutator-line mutagenic systems are based on the mutator (Mu) element, which is one of the most actively proliferating TEs in the maize genome [43]. Mu-like transposons have been identified in many other plant species, including the recently characterized VANDAL transposon family in Arabidopsis [11,44]. Mutator lines exhibit a 50–100 times higher frequency of mutations, as deduced by phenotypic analysis of kernels and seedlings. The “Mutator” locus generated these phenotypic abnormalities was dominant when the lines were crossed with maize lines lacking this locus. A few years later, the causative element for the “Mutator” locus was identified as a transposon and was called Mu1 [45]. Both Mutator and non-mutator maize lines have Mu-like elements, but only the former has Mu elements capable of transposition [46]. Interestingly, it was found that some maize lines carry a Muk locus (‘Mu killer’) that encodes a rearranged version of the autonomus MuDR element that is capable of producing small RNAs that can trigger the DNA methylation of active Mutator elements [47,48]. This natural system consisting of an active TE and its silencer can be useful for the activation of TEs and their heritable silencing when the required number of TEIs and mutagenesis efficiencies are achieved.

Mutator lines have been used to create collections of crop lines that carry TEIs [49,50]. These non-transgenic collections comprising tens of thousands of individuals have been created for maize and rice (TUSC (trait utility system for corn), UniformMu, and MTM) [51,52]. The first maize collection (TUSC) of plants with Mu insertions, developed by Pioneer Hi-Bred, provides useful information for accelerating genotype-based plant breeding and target gene discovery [53]. Transposon tagging has also been applied in rice via the utilization of plants of certain landraces possessing a hyperactive miniature Ping (mPing) transposon [52,54]. For example, the Aikoku and Gimbozu rice cultivars have >1000 mPing insertions compared to 51 copies in the Nipponbare cultivar [52]. Importantly, mPing TEs remained active and produced novel insertions after crossing Gimbozu plants with cultivars without active mPing, paving the way for the generation of mPing insertions in different rice cultivars.

Thus, several crop plant collections have been obtained via the utilization of lines with a natural hyperactivity of distinct TEs. These collections have been used in the plant breeding process and in functional genomic studies via the rapid isolation of genic mutations for reverse genetics.

## 3. Controlled Activation of TEs via PolII and DNA Methylation Inhibition Using Toxins

The above-mentioned methods assume the utilization of mutant plants for TE reactivation. This may significantly hamper the implementation of these methods in the plant breeding process, where the value of a particular genotype is very high. For a long time, it was impossible to reactivate TEs native to a particular plant genotype. However, in 2017, Thieme et al. [20] introduced a novel method called TEgenesis^®^ (patent WO/2017/093317). The addition of zebularine (Z, cytidine analog) and alpha-amanitin (A, a PolII inhibitor) to the plant growth medium led to TE silencing relaxation and the genetic inheritance of novel insertions (Figure 1). Novel insertions were detected in M1 Arabidopsis plants (progeny from plants grown in a ZA-containing medium and subjected to heat stress). However, the utilization of this method in other plant species has not been sufficiently demonstrated. Recently, TEgenesis was applied to reactivate TEs in sunflowers [55]. The authors demonstrated the induction of expression for 16 TE-related loci, and ten of them also showed copy number variation between sunflower varieties, indicating transposition potential. However, the authors did not succeed in growing sunflower plants following TEgenesis. The sunflower seedlings treated with the toxin showed dwarfism, poor stem development, and no lateral roots. Plants grown in the AZ medium did not survive after being transferred from the in vitro culture to the greenhouse. Thus, although TEgenesis is a highly promising method for TE-mediated biological mutagenesis, further optimization is required for its application in crop plants.

## 4. Stress-Mediated TE Transposition Activation

Although DNA methylation is essential for TE silencing, its full erasing from the genome is not sufficient to boost TE activity [56]. The fact that different abiotic and biotic stresses can induce TE expression has been well uncovered in recent reviews [57,58]. It is now becoming clear that both TE-specific and global genomic factors are responsible for TE activation under stressful conditions. TE-specific factors include the existence of a unique motif in LTR sequences that is recognized by transcription factors (TFs) expressed under stress. For example, ONSEN elements possesses a binding site for HSFA1/2, which causes global changes in gene expression and genome organization. Heat stress is one of the most-well-studied stress factors in terms of the activation of TE expression. Attention to TE expression activation by HS was significantly enforced after the discovery of the ONSEN TE family. Stress may activate expression of hundreds of TEs from different families (for example, 363 *A. thaliana* TEs were upregulated under HS [59], but only a small number of them are transposed.

Tissue culture is another stress factor that has received particular attention in the context of the TE transposition activity. In vitro culture may have a two-sided effect on TE transposition. Firstly, similarly to HS, callus may trigger TE activity, and different TEs were shown to be specifically activated during in vitro culture, including LTR retrotransposons Alex1 and Alex3 of carrot [60], Tto of tobacco [61], Tos in rice [62], and ‘Nikita’ in barley [63] (Figure 1). These LTR retrotransposons are active during callus cultivation and generate eccDNAs and new insertions. Secondly, calli may facilitate TE transposition by indirectly influencing TE biogenesis. For example, Masuta et al. [64] demonstrated that ONSEN generates new insertions in calluses but only after HS application. Similar results were observed after the heat treatment of calli from Japanese radish plants [64]. The authors did not find any reliable changes in the expression of RdDM genes or HSFA1 transcription factor nor in the siRNA level or DNA methylation of ONSEN in the callus. Thus, the molecular mechanisms underlying transposon activation in plant tissue cultures remain unclear. Some crop species, such as rice, can be relatively easily regenerated from calli, opening the way for the utilization of TE-based biological mutagenesis. By triggering Tos17 activity in callus, Nipponbare rice collections possessing > 40,000 plants with 5–30 novel Tos17 insertions per plant have been created [65]. Target sequencing of the TEI-flanking regions allowed for the elucidation of insertion biases and provided important information for the further application of this collection in functional genomics and plant breeding. Transposon mutagenesis via callus regeneration has also been successfully applied to another crop, the Japanese radish (*Raphanus sativus*) [64]. The genome of this species contains a heat-inducible ONSEN-like LTR. The authors applied heat stress to the calli and were able to regenerate plants with novel insertions.

Biotic stress, including pathogenic attacks, has also been described as a trigger for TE activity. For example, the ATCOPIA93 (e.g., EVD) family of Arabidopsis LTR retrotransposons can be triggered by biotic stress [66]. A recent study also showed that under flagellin treatment, transposition of the META1 LTR retrotransposon of the Ty1/Copia superfamily is activated [6]. Tnt1 in *Nicotiana tabacum* is also induced by pathogen attacks [27,28,61,67].

Currently, our knowledge of the triggers of plant TE transposition is limited to a few plant species. For future applications of TE-based biological mutagenesis, sophisticated TE-transposition-tracing methods should be developed. While transcriptome-based approaches can be useful for the identification of expressed TEs, the direct detection of TE transposition events at the whole-mobilome scale is a prospective approach. The latest high-throughput methods of mobilome analysis, including circular [68,69,70,71,72] and linear extrachromosomal DNA [73,74,75] sequencing, provide great opportunities for deciphering conditions that are permissive or provocating for TE activity [76,77,78]. Such large-scale screening performed for different developmental stages, stress conditions, and organs would provide valuable information for the further search of TE activity triggers. An alternative approach available for plants with a high number of sequenced genotypes grown under different geoclimatic conditions is a genome-wide association study (GWAS). This method has been successfully applied to show the association between the environment and the activity of certain TE families, revealing TE transposition triggers [6,76].

## 5. Ensuring of Transgenerational Inheritance of Novel TE Insertions

The rate of transgenerational inheritance of novel insertions is an important factor in the utilization of TE-mediated biological mutagenesis. The expression and somatic transposition of a TE do not mean that the insertions of this TE can be transgenerationally inherited. For example, ONSEN elements possess somatic transpositions in response to heat stress, but the inheritance of novel insertions in wild-type plants is very low [17]. Transgenerationally inherited TE insertions must occur in germline cells, gametes, or embryos (GGE cells). Understanding the molecular mechanisms for silencing TEs in GGE cells is crucial for developing TE-mediated biological mutagenesis.

The existence of specific mechanisms for the suppression of transgenerational inheritance of TE insertions in GGE cells was proposed more than 10 years ago and has received considerable attention. Multiple studies have shown that one of the conserved mechanisms protecting GGE cells from TE transpositions is the RdDM pathway that produces siRNAs that cause TE methylation [79]. Interestingly, the latest reports indicate that the RdDM pathway safeguards GGE cells by producing siRNAs that are delivered to GGE cells from surrounding somatic tissues and nurse cells (in trans). This pattern seems to be well-conserved and has been described for different GGE cells, including meiocytes, pollen, eggs, and meristematic cells from SAM (Figure 2).

### 5.1. Tapetum-Derived Small RNAs Silence TEs in Meiocytes

The number of cells that donate siRNA to GGE cells is very different. Long et al. (2021) demonstrated the importance of 24nt tapetum-produced siRNA in determining the silencing state of TE in male meiocytes [80]. These siRNAs are produced in the tapetum and are then delivered to meiocytes when plasmodesmata develop. The 24nt tapetum siRNAs were produced through the canonical RdDM PolIV/RDR2 pathway and required the CLASSY3 (CLS3) chromatin remodeler expressed in tapetum cells (Figure 2A). Interestingly, tapetum-derived siRNAs do not perfectly match their targets in meiocytes, and up to three mismatches are allowed. Tapetal cells can be multinucleated, polyploid, or polytenic [81]. Such genome remodeling may trigger TE expression [82], providing a natural source of TE RNAs for the RdDM machinery. Tapetum-derived siRNAs can also be generated via Pol II transcription of RNA from specific loci, called PHAS [83,84]. siRNAs processed from PHAS RNAs are called phasiRNAs and are 21 or 24nt in length [85,86]. Shorter and longer phasiRNAs accumulate prior to and during meiosis I, respectively. The processing of 21 and 24nt phasiRNAs begins by cleavage of the RNA precursor by 22nt miR2118 microRNA (miRNA) or miR2275, respectively [83,85,86,87]. The functions of 24nt phasiRNA in post-transcriptional gene silencing, gametophyte development, stress response, and plant fertility have been shown [86,88,89]. The low similarity between phasiRNAs and transposons suggests that they are involved in gene transcription regulation rather than TE control [88,90]. Hence, the direct role of phasiRNAs in TE silencing remains unclear.

### 5.2. TE Silencing in Pollen Generative Cell via easiRNAs Movement from Vegetative Cell

The next step in gametophyte development is pollen formation via the meiotic division of meiocytes, resulting in pollen production (Figure 2B). It is well established that components of the RdDM pathway (for example, PolV) are actively involved in pollen development [86]. Mature pollen possesses vegetative and generative cells. siRNA movement from vegetative to generative cells is an important hallmark of pollen development [91]. siRNA movement not only has gene-regulatory functions but also causes TE silencing and prevents their transposition (for example, the Copia95 family in Capsella nrpd1 mutant [92]). The specific class of secondary 21/22nt siRNAs called easiRNAs (epigenetically activated siRNAs) has been shown to move from vegetative to sperm cells [93]. EasiRNAs have been found in DNA-methylation-deficient mutants (e.g., met1 and ddm1) pollen. In contrast to phasiRNAs, which are produced mainly from low-copy sequences, 21/22nt easiRNAs originate from active transposable elements, implying their direct role in TE silencing [93,94]. EasiRNAs are produced in vegetative cells, which have lower DNA methylation levels and a decreased expression of the DDM1 protein. These characteristics suggest that vegetative cells have a “TE-friendly” environment [93]. Similar to phasiRNAs, the production of easiRNAs is initiated by miRNA cleavage [95]. The miRNA-cleaved RNAs are subjected to RDR2- or RDR6-mediated dsRNA conversion followed by easiRNA production by DCL2/4 processing. EasiRNAs are associated with the RdDM pathway via mir845, triggering PolIV easiRNA biogenesis [96]. Thus, pollen is another example of TE silencing in GGE cells via siRNA delivery from nurse cells with direct involvement of the RdDM pathway.

### 5.3. Putative Movement of Small RNAs to Silence TEs in the Egg Cell and Embryo

Small RNA movement during female gametophyte development has been demonstrated recently [97] (Figure 2A). The authors showed that 21nt small RNAs processed from GFP hairpin RNA expressed specifically in synergids can silence the GFP signal in both the egg cell and the central cell. The fusion of the second sperm cell with the central cell results in endosperm development. Embryo and seed development depends heavily on the endosperm. It has been proposed that siRNA can move from the endosperm to embryonic cells. The results obtained on artificial small RNAs of *Arabidopsis* indicated that small RNAs can indeed move from the endosperm to the early embryo, triggering non-cell-autonomous silencing [97] (Figure 2C). However, evidence of whether natural small RNAs can move toward egg cells or embryos is still lacking. Endosperms possess a class of small RNAs called siren (siRNAs in endosperm [98]). These siren small RNAs were detected in ovules, seed coat, embryo, and endosperm, accounting for 90% of all siRNAs in these cells [98]. The siren loci are enriched in some classes of transposons, including class I and Helitrons [98]. This may suggest the role of siren siRNA loci in the protection of embryo cells from TE invasion, although this requires further investigation.

### 5.4. RdDM-Based TE Transposition Silencing in the Shoot Apical Meristem

Alternatively, TE insertions may be inherited if they occur in shoot apical meristem (SAM) cells before they differentiate into flower organs. In most plants, including Arabidopsis, stem cells in the SAM are concentrated in the central zones (CZs), consisting of three cell layers: L1, L2, and L3 [99]. Of these, cells in the L2 layer give rise to meiocytes [100]. The SAM possesses specific RNAi-dependent mechanisms to exclude viruses from stem cells [101]. Based on the known common mechanisms of a plant cell to fight TE activity and viruses, it is logical to propose the existence of a SAM-mediated TE insertion inheritance barrier. How SAM cells protect themselves from TE transpositions and whether there are any specific SAM-TE-silencing pathways are not well understood. Recent studies have shown that GGE cells from the SAM have an additional RdDM-mediated layer of defense against TE transposition. This was demonstrated by population genomics methods [6] and laboratory experiments [20,21,102]. The results of these studies indicate that TE insertions occur in the SAM during the vegetative-to-floral stage transition and before floral organ development. Self-pollination of heat-stressed *nrpd1* plants resulted in homozygote ONSEN insertions in some F1 progeny [102], indicating that the insertions occurred before male and female gametophyte differentiation. Based on these results, it may be concluded that ONSEN insertions occurred during the SAM transition to flowering but before the megaspore mother cells (MMC) and the microspore mother cells (MiMC) were originated. Moreover, the period of ONSEN transposition seems to be very narrow because the individual flowers of *Arabidopsis* plants have flower-specific ONSEN insertions [102]. However, the role of RdDM in TE silencing during flower transition is not well understood. In apples, this stage is characterized by cell reprogramming assisted by 24nt small RNAs of an RdDM pathway origin [103]. Thus, the vegetative-to-floral transition stage is correlated with RdDM pathway activity in apples. The RdDM pathway also regulates the expression of ONSEN in the SAM in response to heat stress, with very low expression levels in wild-type plants and high expression in nrpd1 SAM regions [104,105]. This again indicates the existence of a SAM-specific RdDM-dependent pathway for the control of TE expression and transposition activity [100,106]. Differential expression analysis of SAM cells of *A. thaliana* at different stages compared to non-SAM cells showed that the specific genes of the RdDM pathway were upregulated in the SAM germline (stem) cells, including PolIV subunits NRPD1A, AGO5, AGO9, RNA-dependent RNA polymerases 4 and 5, and histone methyltransferases SUVH4 and SUVR2 [100,107] (Figure 2D). The expression of some TEs, including potentially mobile TEs, is upregulated during the vegetative stage (7 days after germination) [100]. The two detected AGO proteins are specifically expressed in SAM stem cells, and AGO9 is mainly localized in the nucleus of L2 until floral induction [108]. Moreover, AGO5 and AGO9 are associated with TE-derived small RNAs, including 21nt siRNA resembled easiRNAs [108]. These data suggest that AGO5 and AGO9 are SAM stem cell factors that may restrict TE activity during plant development. The authors also demonstrated that long pericentromeric TEs repressed by the DDM1 pathway produce siRNAs in an miRNA845-dependent manner and that these siRNAs are loaded into AGO5 and AGO9. In addition, RdDM pathway (DCL1-3, PolIV, and RDR2) proteins are involved in AGO5-bound small RNA biogenesis. The listed features of SAM small RNAs make them very similar to pollen easiRNAs [108], although whether SAM small RNAs are capable of moving to the SAM from surrounding cells remains to be elucidated.

The above-mentioned examples show that the RdDM pathway, involving transactive tapetum siRNAs or miRNA-directed easiRNAs from pollen and SAM cells, is a major determinant of TE activity in GGE cells. Therefore, these genes are key targets to ensure the inheritance of new TE insertion within the framework of TE-mediated biological mutagenesis (Figure 3A).

## 6. Semi-Random Integration of TEs in Genome

Do TE integration hotspots exist in plant genomes? Several loci that are recurrently attacked by multiple TE insertions have been described in Arabidopsis [6,10], humans [109], and Drosophila melanogaster [110]. However, it should be noted that the recurrent insertions of different TEs into a single locus were observed after the resequencing of natural populations. This means that the observed TEI distribution may be significantly affected by the selection pressure. Recently, long-read sequencing methods have been developed for the detection of somatic insertions [9,111]. The new data from somatic TEI locations suggest that genomic hotspots for TE invasion indeed exist, although the factors (genomic and/or TEs) that direct TEs into them are not yet clear.

The location of TE insertions relative to genes plays a pivotal role in the efficiency of mutagenesis. Additionally, the location of novel TE insertions in a genome often determines how it is subjected to the action of the major drivers of TE elimination from genome recombination, natural selection, and genetic drift [112,113]. The rate of elimination of TE insertion is an important factor in TE-based mutagenesis. Evidence that TE insertions are non-randomly distributed along chromosomes has been accumulating over the last few decades (reviewed by [113,114]). The chromovirus lineage of Ty3/Gypsy LTR retrotransposons is the most-well-studied example in plants. Chromovirus LTR TEs tend to preferentially insert into the pericentromeric regions of chromosomes where the recombination rate is low [113,114]. It was shown that the integrase of chromovirus TEs have a chromodomain that interacts with certain histone marks such as H3K9me2 and H3K9me3, enabling the targeting of heterochromatins [115]. In contrast, some TEs were inserted into the euchromatic regions of the chromosomes. The distribution of the insertions of ONSEN and EVD TEs along Arabidopsis chromosomes is significantly biased toward chromosome arms [6,8,10], with very few insertions being detected in the pericentromeric regions. This was demonstrated by a genomic analysis of genetically inherited TEs [10] as well as by somatic insertion profiling [9]. The TE insertion distribution over gene bodies also exhibits certain biases and differs between different TEs [10,116,117]. TE insertions frequently occur in promoters and in the 5′ and 3′ regions of the genes. However, it is difficult to judge TE insertion preferences based on the analysis of genetically inherited insertions because the observed distribution undergoes polishing by selection pressure. To shed light on the real TE insertion landscape, the detection of somatic TE insertions can provide valuable information.

The biased TE insertion distributions with respect to the major chromosomal landmarks indicate that epigenetic modifications of DNA or histones may influence TE insertion preferences. Indeed, the insertions of VANDAL21 are associated with two histone modifications, H3K4me3 and H3K36me3 [10]. AtENSPM3 and ATCOPIA93, on the other hand, often target genes with H3K27me3 histone marks and histone variants H2A.Z. It has been clearly proven by a couple of studies that ATCOPIA78 (ONSEN elements) and ATCOPIA93 (EVD) insertions are preferentially located in H2A.Z- and H3K27me3-enriched regions [9,10,25]. This is in agreement with the low density of EVD and ONSEN insertions in pericentromeric regions, where the rate of H2A.Z histone and H3K27me3 histone modification deposition is lower than in chromosome arms [118,119,120]. Moreover, mutation in the DDM1 protein, which prevents H2A.Z deposition in the pericentromeric region [118], causes a significant shift in EVD insertion distribution toward the pericentromeric region [9,10]. An unequal distribution of histone variants along the gene body can also explain the frequent occurrence of insertion of some TEs into the TSS [112]. For example, the H2A.Z histone variant is significantly enriched at the 5′ ends and TSS of Arabidopsis genes [118,119,121]. Thus, accumulating data suggest that TEs have certain insertion preferences that are linked to the chromatin state determined by epigenetic factors and histone variant deposition.

Epigenetic-linked TE targeting assumes that the TE insertion site distribution may be linked to the transcription state. Complex bioinformatic analysis of somatic and genetically inherited TE insertions has revealed three main groups of TEs based on their insertion in the gene body (TSS and TTS sites) and the expression intensity of the target genes [112]. The first group (“type-A strategy”) included TEs that were preferentially inserted near the TSS regions of highly expressed genes. Interestingly, the authors showed that usually (20 of 33 insertions) these insertions of the Mu element are not so harmful to gene expression, suggesting that there is less chance for TEs to be purged by evolutionary forces [112]. The insertion frequency at TSS sites suggests that TE proteins involved in the integration mechanism may have an affinity for proteins from PolII-mediated transcription initiation. TEs from the second group (e.g., Ds element of maize) followed the ‘type-B strategy’ and are frequently inserted near both the TSS and TTS sites of the medium-expressed genes. Finally, the third group of TEs with the ‘type-C strategy’ includes TEs (e.g., Tos17 of rice and Spm element of maize) that do not preferentially target either the TSS or TTS sites [112].

Thus, plant transposons have a variety of integration preference scenarios, and in most cases, the factors influencing the integration landscape are not well understood. Based on the results obtained from the plant, yeast, and animal models, it can be concluded that the target site selection for novel TE insertion is determined by several key factors: the host and TE proteins (‘tethering model’ [114]), the chromatin state [10], transcription activity [112], and, very rarely, by the DNA sequence in the integration site [113]. The TE integration process also depends on host proteins, which, together with TE proteins, are involved in target site selection [113,114]. Plant proteins and TE protein features involved in the TE targeting of specific chromatin features of the genome have not been systematically studied in plants. The tropism of TEs toward certain chromatin features means that a TE can change its integration preferences depending on many factors that modulate the distribution of histone variants and epigenetic marks. If this is true, TE-mediated mutagenesis will have a broad range of settings to modify the insertion landscape. The insertion preferences can be linked to transcriptional changes in target genes. For example, ONSEN elements triggered by heat stress have insertion preferences for genes that are downregulated 24 h after stress [9]. It can be proposed that the efficiency and gene-biased preferences of TE-based biological mutagenesis can be modulated by manipulating certain genetic (e.g., DDM1 expression) and environmental (e.g., heat-stress) factors.

## 7. Alternative Methods of DNA Methylation Reduction for Future Use in TE Activation

As discussed above, the existing method of TE-based mutagenesis relies on the implication of plant lines carrying active TEs or on the exploitation of plants carrying mutations in DNA methylation pathways. Alternatively, TEgenesis has been successfully applied to enable TE mutagenesis in Arabidopsis [19,20,21]. This suggests that reducing the activity of genes involved in TE silencing pathways (mainly the RdDM pathway) can be a way to perform TE-based mutagenesis. Knowledge of a gene sequence paves the way for its manipulation. Currently, tremendous success has been achieved in deciphering the whole-genome sequencing of many plant species, including crops with large and complex genomes such as wheat (*Triticum aestivum*, 1C = 16 Gb [122,123]), rye (*Secale cereale*, 1C = 7.9 Gb [124]), and onion (*Allium fistulosum*, 1C = ~12Gb [125]) [126]. Therefore, sequences of TE-silencing genes are readily available. A number of modern techniques allow for the transient silencing of gene expression in plants, including exogenous application of RNA molecules [127] and virus-induced gene silencing (VIGS [128]). These approaches do not require the creation of genetically modified plants but rather temporally suppress the expression of target genes. Another intriguing approach to temporally inhibit gene transcription is CRISPRi (CRISPR-interference), which uses inactive Cas9 (dCas9) and a transcription repression domain or shorter single-guide RNAs [129] that are unable to cut DNA but can bind to it, thus interfering with the transcription process. Recent studies have demonstrated that CRISPR components can be delivered into plant cells using different approaches, omitting the GMO creation procedure [130]. Indeed, a few proof-of-concept studies have reported approaches that can be used to transiently silence DNA methylation genes, including chromomethylase CMT3 [131,132], MET1, and DDM1 [133]. Thus, the available tools may potentially provide a new approach for GMO-free and even DNA-free transposon-mediated biological mutagenesis via the specific manipulation of genes involved in TE silencing.

Many TE-silencing proteins are involved in post-transcriptional gene silencing (PTGS), which also targets dsRNA-producing plant viruses [134]. Viruses and fungi have evolved many anti-silencing proteins known as silencing suppressors (VSRs) that inhibit RNA interference [135]. Some targets of viral anti-silencing proteins are key players in TE silencing. For example, the P19 (VSR from tomato bushy stunt virus) anti-silencing protein binds siRNAs and impairs their loading into AGO1 [136]. Similarly, 2b VSR protein from cucumber mosaic virus (CMV) impairs AGO1 and AGO4 function via direct protein–protein interactions [137,138]. Another example is flagellin (flg22), which may cause DNA demethylation [139]. These proteins can be used as biological agents to activate TEs in plants. Indeed, the transient or permanent overexpression of 2b protein in *A. thaliana* resulted in the transcriptional activation of some TEs [140]. The evolution of anti-silencing proteins is also ongoing. Therefore, TEs can be a source of proteins with demethylation activity. Examples of such TE proteins are growing and include VANC6 and VANC21 proteins of VANDAL DNA transposons of Arabidopsis [131,132], TnpA encoded by the Spm DNA transposon, and MURA protein encoded by the MuDR transposon of maize. The actual number of such anti-silencing proteins encoded by TEs is higher. Importantly, some of these proteins (VANC6 and VANC21) specifically bind to TEs from the corresponding family, allowing for target TE activation in nature. Another protein that counteracts small RNAs is RTL1. Overexpression of RTL1 in Arabidopsis suppresses the production of small RNAs by several Dicer proteins, including DCL2, DCL3, and DCL4 [140]. The proteins listed above provide an interesting direction for the development of novel TE activation platforms.

## 8. Challenges and Emerging Perspectives of TE-Mediated Biological Mutagenesis

Developing methods for TE activation and new TEI inheritances would allow for the artificial acceleration of the natural process of genetic divergence generation. The collection of plants with new TE insertions (iTE plant collections) is a fascinating tool in multiple scientific directions (Figure 3B,C). iTE collections of crop plants can be used directly by breeders as a source of novel phenotypes during plant breeding. Using these collections created for a particular cultivar will allow for the introduction of new genetic variants that reduce the cost and time of the entire plant breeding process. Another obvious application of the iTE collection is functional genomics. TE transposition often leads to significant epigenetic and transcriptomic changes at the site of insertion, thereby increasing the chances of finding an association between insertion and phenotypic changes [21]. Another advantage is the easier process of finding TEIs compared to SNPs. The available methods allow for the target amplification of TEIs without WGS, although primary knowledge of the active TE sequence is required [9,141,142,143,144,145]. iTE populations are also a great source for TE biology investigation as well as for the analysis of epigenetic, genomic, and transcriptomic consequences from TEIs. A recent exploration of hcLines of *A. thaliana* carrying tens of novel ONSEN insertions has shed light on the genomic and transcriptomic effects of TEIs [22]. This study revealed complex and diverse outcomes from ONSEN insertions. It would be interesting to explore the impact of novel insertions on the epigenetic environment at TEI sites. Long-read sequencing followed by cytosine methylation calling could provide a tremendous resolution for answering this question.

Before the broad creation and application of iTE collections, several obstacles must be resolved. First, the developed TEgenesis approaches seem to be very harmful for plants, reducing the plant survival rate [55]; therefore, alternative approaches need to be developed to ensure TEI occurrence in germline plant cells, enabling the genetic inheritance of novel TEIs. Second, the number of active TEs for which the triggering factor is known is extremely low. The establishment of conditions that induce TE activity is crucial for TE-mediated mutagenesis. Third, the stability of novel TEIs in wild-type plants with large genomes (e.g., maize and wheat) remains understudied. Therefore, multigenerational trials are needed for plants carrying novel TEIs. Fourth, even though TE-mediated biological mutagenesis is transgene-free, plants obtained by this method may subject to the GMO regulation rules in some countries [145].

## 9. Conclusions

TE-mediated biological mutagenesis is an emerging tool that has rapidly broadened the genetic and phenotypic diversity of modern crops. However, the number of transgene-free approaches for performing TE mutagenesis in crops is currently limited. Modern knowledge of TE transposition triggers and inhibitors is the foundation for the development of a new generation of TE mutagenesis tools utilizing gene-targeting approaches (for example, VIGS- and CRISPR-based tools). The obtained plant iTE collections will be a great source for plant breeding acceleration as well as for gaining deep insight into the consequences of TE insertion on (epi)genomic, transcriptomic, phenotypic, and ecological levels.

## Figures and Tables

**Figure 1 ijms-24-17054-f001:**
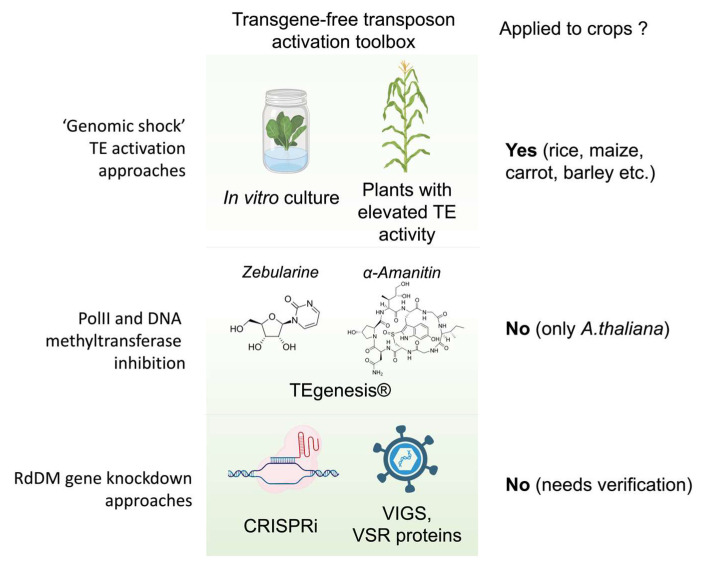
Approaches for TE-mediated biological mutagenesis: application of various stress factors (‘genomic shock’) such as crossing with plant lines possessing hyperactive TEs (e.g., Mutator lines of maize) or plant tissue culture (e.g., activation of Tos17 TE of rice); TEgenesis^®^ (application of stress (e.g., heat stress) to the plants grown on medium containing zebularine (cytidine analog) and alpha-amanitin (a PolII inhibitor)); and new perspective tools (e.g., CRISPRi and VIGS) for transient knock-down of genes involved in RdDM pathway.

**Figure 2 ijms-24-17054-f002:**
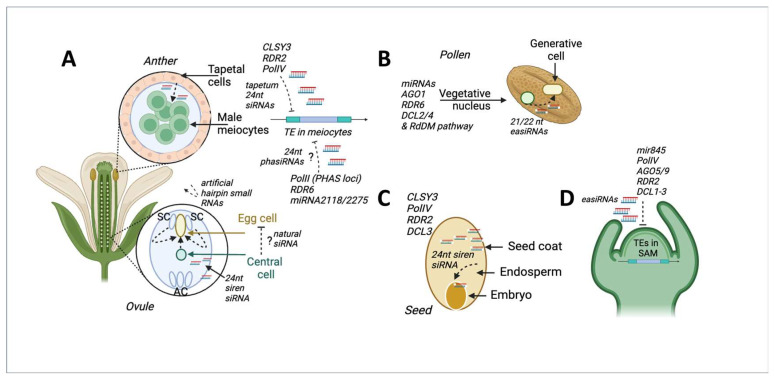
Transposon silencing in germline cells, gametes, or embryos (GGE cells) via small RNA movement (sRNA) from nurse somatic cells. (**A**) TE silencing in meiocytes in anthers and in egg cells in ovules via tapetum-derived sRNAs and central cell/synergid cells (SC) sRNA, respectively. AC—antipodal cells. (**B**) TE silencing in the generative cell of pollen via sRNAs produced by miRNA and RdDM. The figures were generated using BioRender (https://biorender.com/, accessed on 3 June 2023). (**C**) TE silencing via sRNAs movement from endosperm to embryo in a seed; (**D**) easiRNA-mediated TE silencing in the shoot apical meristem (SAM).

**Figure 3 ijms-24-17054-f003:**
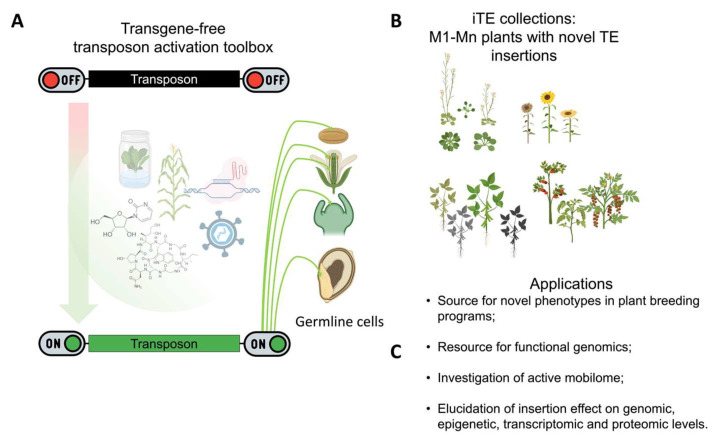
Overview of TE-mediated biological mutagenesis. (**A**) TE activation in the germline cells using different approaches. (**B**) Development of iTE collections (collections of plant lines with novel TE insertions) and (**C**) their applications in research and plant breeding.

## Data Availability

Not applicable.

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
