# Peer review of "Toward Transgene-Free Transposon-Mediated Biological Mutagenesis for Plant Breeding"

_ijms, 2023, doi:10.3390/ijms242317054_

Round 1

Reviewer 1 Report

Comments and Suggestions for Authors

This manuscript finds an interesting field of plant breeding and research. It describes the concept and current knowledge of using TEs to create genetic diversity.  However, the manuscript need some more emphasis on "this technology works", and we have some examples in the list. 

I recommend authors add a table or a paragraph to emphasize what has been achieved in research and breeding with more details. That will make this manuscript more valuable and convincing. Also, listing some traits that had been proven made by new TE actives will also help. 

I also provide some monir editing recommendations:

Lines 92, 114, and 132 italic all Latin names (also still many in the manuscript)

Figure 1, does not explain the TEgenesis® well
Figure 3 could be moved up to Figure 1 because it summarizes the concept of the review.

Words in several lines are cut (ex, line 342). Editing on line-space might be needed.

Figure resolution is low in the manuscript.

Line 145, citations for concept used in the figures. More explanation is also encouraged for the figure.

From Figure 1, there are listed some “applied to crops”. A table listing the applications in crops, with maybe species traits or cultivar output is encouraged. That will make the manuscript more meaningful.

Line 438, try to list some species with a few more details from the reference 127

Author Response

I sincerely appreciate the reviewer's positive assessment of my work as well as their insightful remarks and recommendations. Please see the detailed responses below:

Q1: I recommend authors add a table or a paragraph to emphasize what has been achieved in research and breeding with more details. That will make this manuscript more valuable and convincing. Also, listing some traits that had been proven made by new TE actives will also help. 

A1: Thank you for this comment! I have added new data to exemplify the successful applications of transgene-free TE-based mutagenesis in crops, including maize, rice, and Japanese radish (Raphanus sativus). For these species, large plant collections have been developed, including one created by a breeding company (Pioneer). Unfortunately, it is difficult to trace which cultivars have been created using these collections. I have also added examples of the implications of TE insertions in crop domestication, including tomatoes, B. rapa, and maize, with corresponding examples of agronomic traits.

Q2: Lines 92, 114, and 132 italic all Latin names (also still many in the manuscript)

A2: Corrected throughout the MS.

Q3: Figure 1, does not explain the TEgenesis® well

A3: I have extended the legend of Figure 1.

Q4: Figure 3 can be moved to Figure 1 because it summarizes the concept of the review.

A4: I prefer to leave Figure 3 at the end of the review, as it is also about the perspectives of the iTE line application.

Q5: Figure resolution is low in the manuscript.

A5: Maybe it is because of pdf format. Please check the files that were added separately during submission.

Q6: Line 145, citations for concept used in the figures. More explanation is also encouraged for the figure.

A6: The legend has been significantly modified.

Q7: Line 438, try to list some species with a few more details from the reference 127

A7: I have highlighted some species with large genomes for which genome assemblies have become available.

Reviewer 2 Report

Comments and Suggestions for Authors

This review is about the potential utilization of transposable elements for future crop breeding. Transposons can move in the genome and are therefore the driving force for genetic diversity, which is the basis of breeding. Transposons are widely used to mutagenize plants on a genome-wide scale, e.g. to perform functional genetics studies. But transposon insertions are often detrimental to the host, leading to severe mutant phenotypes. And therefore, host genomes often silence transposons by the deposition of repressive epigenetic marks, small RNAs, easiRNAs, and via RdDM mediated transposition silencing, especially in regenerative tissues. The author summarizes these important mechanisms by refering to recently published studies. There is no question that understanding the molecular mechanisms of transposon regulation is essential to induce its insertion and inheritance. In addition the review highlights mechanisms triggering transposon activiation, such heat stress or biotic stress factors, and it finally gives perspectives for future TE-mediated mutagenesis.

Although this review is very well written and will be of great interest for a broad readership, I have one point of criticism:

There are already some transposon collections in crops, which are mainly used as resources for functional genomics. Very often transposition of mobile elements leads to disadvantages for the plants, e.g. when the transposon jumps in an exon of an „essential“ gene. Although it is not mandatory to name these transposon collections, the author could emphasize a bit more in detail that transposon re-activated insertion often leads to detrimental effects to the host plants. Hence, transposon re-activiations seem to show a limited usability to breed „better“ plants. To solve that issue, the author could easily highlight some examples of insertions that showed positive effects: e.g. the devolpment of teosinte to maize. Teosinte is highly branched whereas maize has only one stem. Reduced branching is caused by a mobile element insertion in the tb1 gene. Another idea to solve that issue would be, to discuss whether the already available transposon collections are already used in breeding programs.

Minor points:

Line 86:               Add reference 44 here at the end of the sentences, because the author referred to this publication.

Line 125:             explain in a bit more detail what „genomic shock“ means.

Line 128:             Replace „Robertsonian´s“ by „Robertson`s“

Line 500:             Refer to Figure C: i.e. replace „and their applications“ by „and (C) their applications“

Author Response

I sincerely appreciate the reviewer's positive assessment of my work as well as their insightful remarks and recommendations. Please see the detailed responses below:

Q1: There are already some transposon collections in crops, which are mainly used as resources for functional genomics. Very often transposition of mobile elements leads to disadvantages for the plants, e.g. when the transposon jumps in an exon of an „essential“ gene. Although it is not mandatory to name these transposon collections, the author could emphasize a bit more in detail that transposon re-activated insertion often leads to detrimental effects to the host plants. Hence, transposon re-activiations seem to show a limited usability to breed „better“ plants. To solve that issue, the author could easily highlight some examples of insertions that showed positive effects: e.g. the devolpment of teosinte to maize. Teosinte is highly branched whereas maize has only one stem. Reduced branching is caused by a mobile element insertion in the tb1 gene. Another idea to solve that issue would be, to discuss whether the already available transposon collections are already used in breeding programs.

A1: Thank you for this comment. We have added new data to exemplify the successful application of transgene-free TE-based mutagenesis in crops including maize, rice, and Japanese radish (Raphanus sativus). For these species, large plant collections have been developed, including one created by a breeding company (Pioneer). Unfortunately, it is difficult to trace which cultivars have been created using these collections. I have also added examples of the implications of TE insertions in crop domestication, including tomato, B. rapa, and maize.

Q2: Line 86:               Add reference 44 here at the end of the sentences, because the author referred to this publication.

A2: The reference has been added.

Q3: Line 125:             explain in a bit more detail what „genomic shock“ means.

A3: I realized that the „genomic shock“ is not appropriately used here. I have removed this from the paragraph and added an extended legend for Figure 1.

Q4: Line 128:             Replace „Robertsonian´s“ by „Robertson`s“

A4: Corrected.

Q5: Line 500:             Refer to Figure C: i.e. replace „and their applications“ by „and (C) their applications“

A5: Corrected.

Reviewer 3 Report

Comments and Suggestions for Authors

This review on TE-mediated mutagenesis and its applications on plant breeding summarized some historical findings and recent updates on this specific topic. Overall, it has merits in drawing attention to this promising approach for crop development. I have the following comments and questions that can be addressed to enhance the review manuscript.

Line 52-53 I find this statement to be somewhat overstated. TE may serve as one of the main sources.

Line 54, 64, 78 Mutagens are usually defined as agents that cause mutations. TEs themselves cannot be considered as mutagens.

Line 65-66 What are these TE mechanisms? What are the most recent molecular tools indicated here? Please briefly describe.

Line 73, 92 All species names should be italicized.

Line 68-77 The sentences in this paragraph are somewhat redundant. Please aim for conciseness and write succinctly.

Line 95 Please use the italic “in vitro”.

Line 97 GMO legislation in some countries doesn’t necessarily mean the use of genetic modification approaches is not possible in plant breeding research. Instead, it primarily regulates the release and commercialization of modified varieties.

Line117 The figure is not consistent with this statement. Instead, the figure shows that TE activation has been applied to crops. To which category does the application of epiRILs belong?

Line 147-149 It is a bit hard to follow the logic here. Please clarify.

Line 166 Why isn’t this approach demonstrated in Figure 1?

Line 214 I suggest reiterating certain points from this section, such as “transgenerational inheritance of TE transposition”, under the “challenges of TE-mediated mutagenesis” discussed later in the manuscript.

Line 498 “Investigation of active mobilome” as one of the applications is not described in the text.

Line 520 change it to “… (epi)genomic, transcriptomic, phenotypic and ecological levels”.

Comments on the Quality of English Language

The English language usage meets the standard, maintaining a generally readable expression throughout. However, some layout, sentence structure or word choice could be improved for conciseness and clarity.

Author Response

I sincerely appreciate the reviewer's positive assessment of my work as well as their insightful remarks and recommendations. Please see the detailed responses below:

Q1: Line 52-53 I find this statement to be somewhat overstated. TE may serve as one of the main sources.

A1: Corrected.

Q2: Line 54, 64, 78 Mutagens are usually defined as agents that cause mutations. TEs themselves cannot be considered as mutagens.

A2: Mutagens are indeed agents that cause mutations. For example, chemical mutagenesis uses chemical mutagens, such as EMS. Here, I use the previously introduced term ‘transposon mutagenesis, ’ where transposons are the primary agents causing mutations. Therefore, they are called mutagens. This usage is actually also in concordance with previous reports (e.g. https://doi.org/10.1093/genetics/165.1.243, https://doi.org/10.1080/713610849 ) .

Q3: Line 65-66 What are these TE mechanisms? What are the most recent molecular tools indicated here? Please briefly describe.

A3: These mechanisms and tools are described in the main text of the manuscript. However, I agree that some examples should be mentioned in the Introduction. I have added them.  

Q4: Line 73, 92 All species names should be italicized.

A4: Thank you. It was corrected.

Q5: Line 68-77 The sentences in this paragraph are somewhat redundant. Please aim for conciseness and write succinctly.

A5: This paragraph has been modified accordingly.

Q6: Line 95 Please use the italic “in vitro”.

A6: Corrected.

Q7: Line 97 GMO legislation in some countries doesn’t necessarily mean the use of genetic modification approaches is not possible in plant breeding research. Instead, it primarily regulates the release and commercialization of modified varieties.

A7: Thank you for this comment. This sentence has been revised accordingly.

Q8: Line117 The figure is not consistent with this statement. Instead, the figure shows that TE activation has been applied to crops. To which category does the application of epiRILs belong?

A8: epiRILs are self-pollinated plants carrying mutations in DNA methylation-controlling genes (for example MET1 or DDM1). This should be classified as a ‘genomic shock.’ However, epiRILs have not been used for TE activation in crops and are not included in Figure 1.  

Q9: Line 147-149 It is a bit hard to follow the logic here. Please clarify.

A9: I agree. I have corrected this sentence accordingly.

Q10: Line 166 Why isn’t this approach demonstrated in Figure 1?

A10: Stress-mediated TE activation (‘in vitro culture’) is shown in Figure 1.

Q11: Line 214 I suggest reiterating certain points from this section, such as “transgenerational inheritance of TE transposition”, under the “challenges of TE-mediated mutagenesis” discussed later in the manuscript.

A11: I would prefer to leave the structure as it is now because the last section of MS only summarizes the main challenges of TE-mediated mutagenesis.

Q12: Line 498 “Investigation of active mobilome” as one of the applications is not described in the text.

A12: The paragraph briefly summarizing the modern method for genome wide mobilome investigation is in the section “4. Stressed mediated TE transposition activation”

Q13: Line 520 change it to “… (epi)genomic, transcriptomic, phenotypic and ecological levels”.

A13: Thank you! It was modified.

Round 2

Reviewer 1 Report

Comments and Suggestions for Authors

I think it's ready to go. The photo resolution needs to be improved, I think the editor team needs to help it.